

# The low survival rate of European hare leverets in arable farmland: evidence from the predation experiment

Jan Cukor[1,2], Jan Riegert[3], Aleksandra Krivopalova[2], Zdeněk Vacek[2] and Martin Šálek[1,4,5]

[1] Forestry and Game Management Research Institute, Prague, Czech Republic
[2] Faculty of Forestry and Wood Sciences, Czech University of Life Sciences Prague, Prague, Czech Republic
[3] Department of Zoology, Faculty of Science, University of South Bohemia, České Budějovice, Czech Republic
[4] Czech Academy of Sciences, Institute of Vertebrate Biology, Brno, Czech Republic
[5] Faculty of Environmental Sciences, Czech University of Life Sciences Prague, Prague, Czech Republic

## ABSTRACT

The low survival rate of leverets may significantly contribute to steep population declines and slow recovery of European hares (*Lepus europaeus*). However, the leveret survival rate in farmlands with different landscape structures is poorly understood, and the existing evidence comes mainly from Western Europe. In this study, we explored the survival of leveret hare dummies along linear semi-natural habitats in homogeneous Central European arable farmland during the main part of the European hare reproduction period (March–April) in 2019 and 2020. The survival rate of hare leverets during the 14-day period was only 22.2%, and all predation events were recorded during the first six days of the experiment. Mammalian predators were responsible for 53.1% of predation events, avian predators for 40.8%, and agricultural operations for 6.1%. The red fox (*Vulpes vulpes*) was the dominant predator in our study area and was the primary cause of leveret dummy mortality (32.7%), but it also had the highest use-intensity and visit frequency of all of the study plots. Predation by avian predators was associated with patches of lower vegetation height and cover (such as plowed fields) and during daylight hours, whereas the opposite was true for mammalian predators. We propose that improving the habitat quality of arable landscapes by increasing the proportion and quality of extensively used non-farmed habitats (*e.g.*, set-asides, wildflower areas, extensive meadows, fallow land, and semi-natural habitats on arable land) providing cover and shelter for leverets could be an effective management measure for reducing predation risk on leverets.

# INTRODUCTION

The intensification of farming management during the last century, particularly since post-World War II, resulted in widespread homogenization of the formerly mosaic-like and heterogeneous agricultural landscape with profound negative effects on farmland

Corresponding author
Martin Šálek, martin.sali@post.cz

biodiversity (*Benton, Vickery & Wilson, 2003*; *Stanton, Morrissey & Clark, 2018*). Large-scale degradation and loss of semi-natural habitats that are often surrounded by an intensively used agricultural matrix may lead to farmland species population declines (*Benton, Vickery & Wilson, 2003*), partially due to increasing predation pressure along the remaining semi-natural habitats and their edges (*Evans, 2004*; *Šálek et al., 2010*). Similarly, increased predation pressure in agroecosystems degraded by intensive farming may be linked with increased numbers of avian and mammalian generalist predators that adapted to or even benefited from agricultural development (*Evans, 2004*). Therefore, agricultural intensification may increase the negative effect of predation on the survival and reproductive performance of declining farmland species (*e.g.*, *Brickle et al., 2000*; *Evans, 2004*; *Panek, 2005*; *McMahon et al., 2020*).

The European hare (*Lepus europaeus*) is a farmland specialist mammal that has experienced a steep population decline during the last few decades leading to conservation concerns in many regions in Europe (*Smith, Jennings & Harris, 2005*). The reduction of landscape heterogeneity and semi-natural habitats, associated with the specialization of agricultural production (linked with a decrease in crop diversity), are the ultimate drivers of its long-term population decline (*Smith, Jennings & Harris, 2005*). However, other factors, such as low leveret survival, may also play an important role in population reduction. In particular, previous evidence has shown that leveret survival is extremely low in farmlands with different landscape structures, preventing population growth and recovery (*Voigt & Siebert, 2020*; *Karp & Gehr, 2020*). For example, previous radiotelemetry studies have shown that hare leveret survival is 33% within the first four weeks of their life in Germany (*Voigt & Siebert, 2020*) and only 18% at the end of the first month of life in Switzerland (*Karp & Gehr, 2020*). However, the critical lack of realized radiotelemetry studies on leveret mortality is the missing evidence of a predator structure. Furthermore, low survival of hare leverets may be even more pronounced in structurally simple arable-dominated farmlands with a low proportion of extensively used non-farmed arable land (*e.g.*, set-asides, wildflower areas, extensive meadows, fallow land, and semi-natural habitats), which may represent crucial high-quality habitats for European hares (*Petrovan, Ward & Wheeler, 2013*; *Meichtry-Stier et al., 2014*; *Weber, Roth & Kohli, 2019*; *Schai-Braun et al., 2020*). However, they may also act as ecological traps due to increased predator activity within these habitats (*Červinka et al., 2013*). Surprisingly, despite leveret survival being a critical component of population dynamics (*Schai-Braun et al., 2020*), there is a general lack of research on this topic, especially in Central European farmlands, where the species significantly declined during the last few decades (*Sliwinski et al., 2019*).

Therefore, the main aim of this study was to estimate European hare leveret survival by evaluating the predation rate on dummies mimicking European hare leverets within intensively used arable farmland. In particular, the specific aims of our study were (i) to identify individual predator species and assess the survival rate for exposed hare dummies, (ii) to assess use-intensity and visit frequency of individual predator species, (iii) to evaluate the effect of local habitat characteristics (*i.e.,* vegetation structure) on predation rates by mammalian and avian predators, and finally, (iv) to compare the predation times of avian

and mammalian predators during the day, as searching patterns and daily activity of avian and mammalian predators may differ substantially.

## MATERIALS & METHODS

### Study area and data collection

The research was conducted in an intensively used arable-dominated agricultural landscape in Northern and Central Bohemia, Czech Republic (Fig. S1). The study region is located within a flat lowland landscape (altitude range 220–260 m a.s.l.) characterized by large arable fields with an average size > 20 ha, which is typical for conventionally managed agricultural land in the Czech Republic (*European Communities, 2008*). The fields are primarily used for the cultivation of cereals, oilseed rape, maize, and sugar beet with low representation (<5%) of extensively used non-farmed arable land, predominantly represented by linear woody or grassy strips, shrub or forest patches, ditches, or extensive grasslands.

The survival rate evaluation was performed using simulated dummies mimicking European hare leverets. These hare dummies were made from 15 × 5 cm hare skin and filled with hay to resemble hare leverets with a weight of ca. 200 g, corresponding to the age of ca. ten days (Fig. S2). Moreover, we sprinkled the immediate surroundings of each dummy (2-m buffer) with 5 ml of domestic rabbit urine to mimic the leveret scent (see also *Šálek et al., 2010*). The dummies were fixed into the ground with thin strings linked with wooden sticks located underneath them, as some attacks of avian predators may be too fast to identify the predator to species level according to the camera trap trigger speed (see below). In total, the experiment was accomplished within 14 consecutive days (*i.e.,* each leveret hare dummy was monitored for 14 days in the field).

The dummies were placed along linear woody and grassy strips (width 2–12 m) within the agricultural landscape, representing the most common in-field semi-natural vegetation inside the study area. More specifically, the European hare leveret dummies were installed at a 3-m distance from the hedge into the crop fields (spring cereals, winter cereals, plowed field, and stubble field), as the majority of hare leveret activity is situated in the narrow edge zone (*Voigt & Siebert, 2019*). Locations of individual study plots were randomly selected (random treatment selection) before the fieldwork using detailed and recent satellite pictures utilizing GIS tools under two conditions: (i) individual study plots were located at least 500 m from each other to achieve the spatial independence of the sites and (ii) study plots were situated at least 500 m from human settlements. Overall, we monitored 48 independent study plots between the beginning of March through mid-April 2019 (30 unique study plots) and 2020 (28 other unique study plots). The chosen time period corresponds with the peak of the European hare reproduction season in Europe (*Lincoln, 1974*; *Broekhuizen & Maaskamp, 1981*).

Predation events and identification of individual predators were monitored using UO Vision UV 595 HD camera traps with an invisible IR camera, trigger speed of 0.65 s, and HD video recording (*Cukor et al., 2021*). Camera traps were installed 3–5 m into shrub or tree vegetation to visually cover the dummies and their surroundings. The predation

event was considered only when predators pecked, clawed, or bit into the dummy or were sniffing above it. For the subsequent analyses, we divided the predation events into two types: (i) primary predation representing the first recorded predation of the dummy captured on the camera traps, and (ii) repeated predation showing repeated events on the same dummy during the 14-day study period. Primary predation was used to evaluate the survival rate curve, whereas repeated predation was used for the analyses of the predation events sequence (*i.e.*, order of predator events during a 14-day period; see also *Cukor et al., 2021*) and predation during the day.

To evaluate the effect of local habitat characteristics on the predation rate and predator composition, we measured the vegetation structure in the immediate surrounding area of the dummies, which could play a crucial role in the predation rate and dummy detection (*Crabtree, Broome & Wolfe, 1989*; *Bellamy et al., 2018*). In particular, we measured vegetation height (average height of vegetation in cm) and visually estimated vegetation cover (% of ground covered with vegetation) in a 3 × 3 m square around the dummy. The vegetation structure was measured on the first day of the experiment by the same person (JC) to avoid inter-observer bias. We did not evaluate the effect of landscape and regional characteristics on the predation rate due to the small sample size for individual predator species.

The experiment took place under the animal care approval n. 63479/2016-MZE-17214 (Institutional Animal Care; Ministry of Agriculture of Czech Republic). The study was accomplished following the relevant national and international guidelines regarding animal welfare. The monitoring was performed with camera traps, so no animal species was stressed or disturbed within the experiment.

## Data analysis

We used the Kaplan–Meier method (*Kaplan & Meier, 1958*) to fit the survival rates of exposed hare dummies using Statistica 14 (*TIBCO Software Inc., 2020*). For this analysis of survival rates, we used only the first predation events with the number of days to the predation event as the independent variable ($n = 46$). This method enabled us to present the descending pattern of survival rate with the number of days after the start of the experiment (*Dudley, Wickham & Coombs, 2016*).

To compare visitation rates among individual predators, we calculated for each predator species the number of its visits during the 14-day period (*e.g.*, use-intensity) and the percentage of days with species presence within a 14-day period (*e.g.*, visit frequency) after *Zitzmann & Reich (2022)*.

We performed GLMM analysis with Template Model Builder (TMB) using the glmmTMB function (package glmmTMB; *Brooks et al., 2017*) in R 4.0.5 software (*R Core Team, 2021*). This analysis was used because of the frequent occurrence of zero values in the dependent variable. We tested the effect of interactions of independent variables (daytime (hour), vegetation cover (%), vegetation height (cm), and crop type (spring cereals, winter cereals, plowed field, and stubble field)) with the predator type (mammal/bird) on the dependent variable predation event (yes/no). We used the event order at each station as a variable with a random effect. The dependent variable had a binomial distribution, and

we used the Logit link function. Since the time values are on a circular scale similar to the directions of animal movements (*Bastchelet, 1981*), we used a transformed variable as recommended by *Cremers & Klugkist (2018)*. We calculated two independent values ($x$-hour and $y$-hour) for each time based on the angle on a circular scale, and we used their interaction in the analysis. Calculations were done according to the following equations: $x$-hour $= \sin (2\pi * (\text{time}/24))$, $y$-hour $= \cos (2\pi * (\text{time}/24))$. First, we built a null model without independent variables, and then we compared this null model with each alternative model using anova function in R 4.0.5 software (*R Core Team, 2021*). After testing the effect of interactions between predator and crop type, we used function lsmeans (package lsmeans) in R 4.0.5. software to perform *post-hoc* tests (*Lenth, 2016*). For all the alternative models, we checked their homoscedasticity (*i.e.,* constant variance) by plotting residuals and fitted values using the ggplot2 function in R 4.0.5 (*Wickham, 2016*). The fitted curves were, in all cases, around zero and not funnel-shaped.

## RESULTS

In the 46 study plots, we observed 48 first-time predation events and 127 total predation events. Survival rate of dummies was low with 78% of dummies being predated within the first six days. Surprisingly, the rest of the dummies that overcame this period (*i.e.,* the first six days) were not depredated. We recorded a steep drop in the survival rate after the first two days following exposure and then the survival rate was relatively stable until the sixth day after exposure (Fig. 1). Mammalian predators were responsible for 53.1% of overall predation, followed by avian predators (40.8%). Agricultural operations were responsible for 6.1%. The red fox (*Vulpes vulpes*) and common buzzard (*Buteo buteo*) were the dominant predators and were responsible for 32.7% and 18.4% of predation events, respectively (Table 1). The highest use-intensity was found for red fox and domestic cat (*Felis catus*), and the highest visit frequency was recorded, again for red fox, followed by European badger (*Meles meles*) and domestic cat (Table 1). The sequence of predation events showed that the first predation mainly occurred with red foxes and common buzzards, and the same was true for the second-order predations (Fig. S3).

When we tested the effect of environmental variables on the presence/absence of predation events ($n = 127$), we found that most of the variability was explained by the interaction between the crop type and predator type (GLMM analysis, d.f. = 9, 16.7% of explained variability, correlation of fixed effects = 0.68, Chi = 27.44, $P < 0.001$). Using *post-hoc* tests, we further found that only the comparison of the distribution of presence/absence of predation events within the plowed field between mammal and bird predators was statistically significant (estimate = 3.48, z-ratio = 2.97, $P = 0.048$). The number of predation events made by bird predators was higher compared to mammal predators (Fig. 2D). The effect of the interaction of predator type and vegetation cover was also statistically significant (GLMM analysis, d.f. = 5, 14.4% of explained variability, correlation of fixed effects = 0.64, Chi = 23.71, $P < 0.001$). Bird predators attacked the hare dummies when the vegetation cover was low, and the opposite was true for mammal predators (Fig. 2A). Similarly, the effect of interaction between predator type

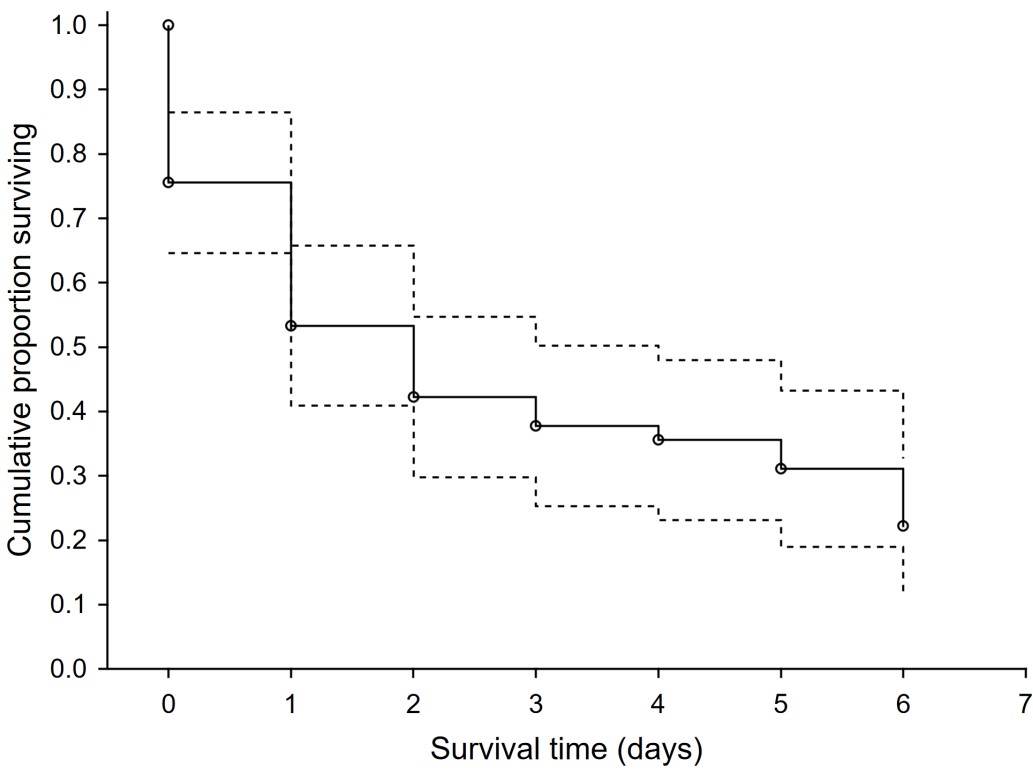

**Figure 1** **The survival rate function for exposed hare dummies during the time after the exposure.**
Kaplan-Meier fitting method ($n = 46$ predation events). Dashed line—95% lower and upper confidence limits.

and vegetation height was also statistically significant (GLMM analysis, d.f. = 5, 14.2% of explained variability, correlation of fixed effects = 0.66, Chi = 23.44, $P < 0.001$). The bird predators attacked the hare dummies when the vegetation was short, and the opposite was true for mammal predators (Fig. 2B). We further found that predation by mammals occurred mainly during the night, while attacks by birds occurred mainly during the day (GLMM analysis, d.f. = 9, 13.6% of explained variability, correlation of fixed effects = 0.96, Chi = 22.43, $P < 0.001$, Fig. 2C).

## DISCUSSION

Our study illustrates the first evaluation of the survival rate of dummies mimicking European hare leverets in Central Europe. The farmland landscape in the Czech Republic has been crucially modified by landscape homogenization and currently is represented by some of the largest arable blocks across Europe (*European Communities, 2008*; *Šálek et al., 2021*). Based on our predation experiment, the survival rate of European hare leverets during the 14-day period was low (22.2%), which is comparable with previous studies based on radiotelemetry of leverets. In particular, a study from Germany showed a 33% leveret survival in the first four weeks of their life (*Voigt & Siebert, 2020*) and only an 18% survival rate at the end of the first month of life, according to a study from Switzerland (*Karp*

**Table 1** Intensity use (number of visits during the 14-day period), visit frequency (percentage of days with species presence within a 14-day period), and number (%) of predation visits and events attempts for all predator species and agricultural operations.

| Species/human activity | Intensity use (mean ±s.d.) | Visit frequency (mean ±s.d.) | Number of all visits (%) | Number of predation events (%) |
|---|---|---|---|---|
| Agricultural operations | 0.06 ±0.24 | 0.46 ±1.75 | 3 (2.3) | 3 (6.1) |
| *Buteo buteo* | 0.21 ±0.41 | 1.52 ±2.92 | 10 (7.7) | 9 (18.4) |
| *Canis lupus f. familiaris* | 0.28 ±0.61 | 1.98 ±4.35 | 13 (10.0) | 4 (8.2) |
| *Carnivora* | 0.02 ±0.14 | 0.15 ±1.03 | 1 (0.8) | 0 (0.0) |
| *Circus aeruginosus* | 0.02 ±0.14 | 0.15 ±1.03 | 1 (0.8) | 1 (2.0) |
| *Corvus corax* | 0.09 ±0.40 | 0.61 ±2.88 | 4 (3.1) | 2 (4.1) |
| *Corvus corone* | 0.09 ±0.35 | 0.61 ±2.48 | 4 (3.1) | 3 (6.1) |
| *Felis catus* | 0.38 ±0.98 | 2.74 ±7.00 | 18 (13.8) | 1 (2.0) |
| *Garrulus glandarius* | 0.06 ±0.24 | 0.46 ±1.75 | 3 (2.3) | 2 (4.1) |
| *Martes* sp. | 0.11 ±0.31 | 0.76 ±2.20 | 5 (3.8) | 3 (6.1) |
| *Meles meles* | 0.36 ±1.04 | 2.58 ±7.43 | 17 (13.1) | 2 (4.1) |
| *Mustela putorius* | 0.02 ±0.14 | 0.15 ±1.03 | 1 (0.8) | 0 (0.0) |
| *Pica pica* | 0.11 ±0.37 | 0.76 ±2.65 | 5 (3.8) | 3 (6.1) |
| *Sus scrofa* | 0.06 ±0.32 | 0.46 ±2.28 | 3 (2.3) | 0 (0.0) |
| *Vulpes vulpes* | 0.89 ±1.43 | 6.38 ±10.23 | 42 (32.3) | 16 (32.7) |

*& Gehr, 2020*). Similarly, the survival rate seems to be higher (range: 35–51%) for other lagomorph species, such as Snowshoe hare (*Lepus americanus*), Tehuantepec jackrabbit (*Lepus flavigularis*), or Pygmy rabbit (*Brachylagus idahoensis*) (see review in *Karp & Gehr, 2020*). Still, our predation experiment may not reflect the actual rate of natural predation (for more details, see Study Limitations) and, therefore, further studies, chiefly based on radiotelemetry, are crucial to evaluate the real predation pressure on leverets in farmlands with contrasting landscape patterns.

Interestingly, all predation events were recorded during the first six days of the experiment. There can be several explanations for the increased predation within the first days of the experiment. Firstly, the predator foraging activity and scanning behavior are primarily concentrated on suitable patches (such as habitat edges, see below) within their home ranges, and therefore, within the first days of the experiment, there might be a high probability of detection of installed hare leveret dummies. Secondly, scent attractant (*i.e.,* domestic rabbit urine; see Materials & Methods) used for mimicking leveret scent at the beginning of the experiment may fade away, which may result in low detectability of leveret dummies by mammalian predators. Thirdly, the decrease of predation in time may be linked with the progress of the vegetation season as a result of higher dummy concealment due to higher vegetation cover and the height of growing crops and weedy plants, which may be important, especially for avian predators. However, the vegetation growth within a 14-day period (the length of our experiment) is relatively short to explain the reduced predation rate within the time of our experiment.

Recorded mammalian and avian predators belong to an extensive and numerous generalist predators inhabiting open agricultural landscapes (except for domestic cats and

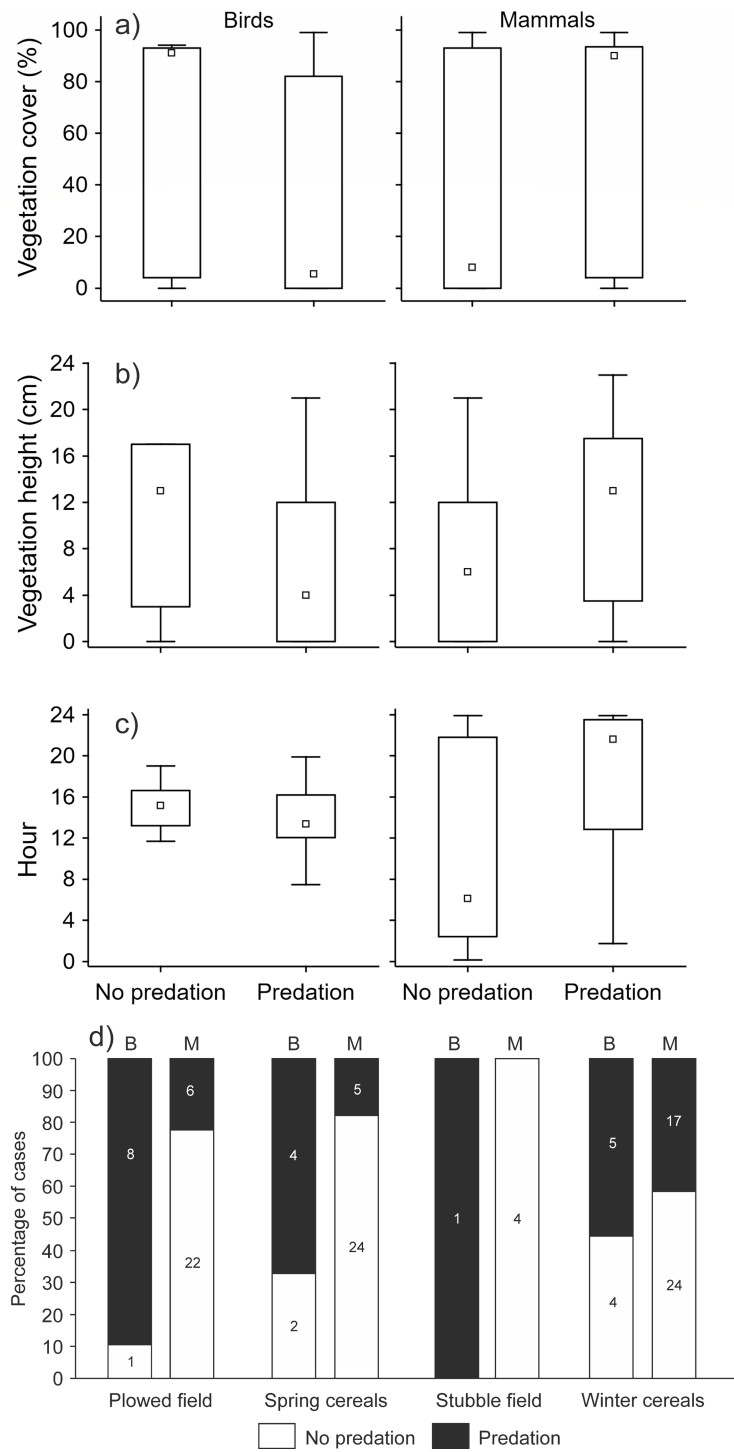

**Figure 2** **The effect of interaction between predator type (mammal, bird) and (A) vegetation cover, (B) vegetation height, (C) daytime, and (D) crop type on the presence/absence of predation events.** GLMM analyses with order of event within a station as the variable with random effect ($n = 127$). Small square–median, box–25–75% of data, whiskers–non-outlier range.

dogs that are associated with human settlements; *Červinka et al., 2013*; *Šťastný et al., 2021*). Populations of several generalist predators (*e.g.*, red fox, stone marten *Martes foina*, raven *Corvus corax*, Eurasian jay *Garrulus glandarius*, red kite *Milvus milvus*, magpie *Pica pica,* or wild boar *Sus scrofa* markedly increased during the last several decades in Central Europe (*Anděra & Hanzal, 1996*; *Jerzak, 2001*; *Goldyn et al., 2003*; *Massei et al., 2015*; *Šťastný et al., 2021*)). This is particularly true for the red fox (*Chautan, Pontier & Artois, 2000*; *Delcourt et al., 2022*), which was responsible for the majority of predation events, and was the most dominant recorded predator at individual study plots, steeply increased in various agroecosystems. It is considered as the main predator of the hare leverets (*Reynolds & Tapper, 1995*) and ground-nesting farmland birds (*Panek, 2005*; *Roos et al., 2018*; *McMahon et al., 2020*). Recorded mammalian species, and red fox in particular, utilize a variety of different habitat types. However, their foraging activity is mainly focused on edge habitats due to increased prey densities or using edges as travel lines in otherwise hostile farmland, or both (*Šálek et al., 2009*; *Šálek et al., 2010*; *Červinka et al., 2013*), resulting in increased predation pressure on hare leverets and other ground-nesting birds along habitat edges (*Evans, 2004*).

Vegetation structure and subsequent extent of leveret concealment within habitats may play a crucial role in the predation rate on a local scale. However, the predation may depend on differences in the search pattern of avian and mammalian predators that rely on different clues when searching for prey. In general, mammalian predators mostly rely on chemical cues, whereas avian predators depend on visual detection (*Apfelbach et al., 2005*). Therefore, dense cover around the dummies might affect the predation rate, especially by avian predators. In line with this, we found that vegetation cover and height were essential factors in the predation rate for avian predators, with increased predation recorded for plots with lower cover and vegetation height (see also *Crabtree, Broome & Wolfe, 1989*; *Bellamy et al., 2018*). Similarly, in contrast to mammalian predators, the predation of avian predators was significantly higher at the plowed field, *i.e.,* habitats composed of bare ground or sparse vegetation. Finally, we found that predation caused by mammalian predators was primarily recorded during night hours, whereas avian predators during the day, which generally coincides with the activity patterns of nocturnally active mammalian predators and diurnal avian predators (see also *Ashby, 1972*; *Cukor et al., 2021*).

## STUDY LIMITATIONS

We are aware of two serious limitations of our research. First, our experiment was focused on estimating survival rates based on simulated leveret dummies, which may not reflect the actual rate of natural predation, as demonstrated by previous comparisons of predation rates of natural and artificial nests (*Buler & Hamilton, 2000*). Still, predation experiments with dummies (or simulated nests) may help to reveal predation risks and to determine dominant predators (*Fernex, Nagel & Weber, 2011*), which is not possible to determine by previous radiotelemetry studies on European hare leveret mortality (*Voigt & Siebert, 2020*; *Karp & Gehr, 2020*). Second, the predation experiment was done along linear habitats associated with increased predator activity and, thus, higher predation risk

for leverets (*Fernex, Nagel & Weber, 2011*). This might overestimate predation pressure, as hare leverets may be more uniformly spaced across the landscape and various(cropped and non-cropped) habitats. However, *Voigt & Siebert (2019)* found that within the intensively used agricultural landscape, the majority of newborn leverets were found close to habitat edges, and leverets up to the fifth week of life spent the majority of daytime and nighttime in the narrow edge zone (up to 20 m from the habitat edge). Therefore, our results of survival rate should be treated cautiously. Yet, it still brings essential insight into the critical phase of the European hare lifecycle, which may have important implications for species conservation and landscape management.

## CONSERVATION IMPLICATIONS

In conclusion, the low survival rate of the European hare could be an important cause of the decline of the European hare population and ground-nesting farmland birds (*Evans, 2004*; *Panek, 2005*; *McMahon et al., 2020*) and may be driven by anthropogenic simplification of the farmland ecosystems. Landscape simplification *via* agricultural intensification may lead to higher predation risk due to increased densities of generalist (avian and mammalian) predators and their concentration on remaining species-rich habitat edges. Moreover, we demonstrated that predation by birds was associated with patches composed of lower vegetation height and cover, whereas the opposite was true for mammalian predators. Therefore, landscape management focusing on habitat restoration of high-quality, extensively used habitats within farmland, provides enhanced canopy cover/shelter (*e.g.*, dense herb and shrub vegetation) for leverets and adults and might reduce predation risk from avian predators (*Pépin & Angibault, 2007*; *Neumann et al., 2011*; *Šálek et al., 2023*). This can be achieved by the higher implementation of set-asides, wildflower areas, fallow land, and semi-natural habitats on arable land (*Petrovan, Ward & Wheeler, 2013*; *Meichtry-Stier et al., 2014*; *Pavliska et al., 2018*; *Weber, Roth & Kohli, 2019*; *Schai-Braun et al., 2020*; *Šálek et al., 2022*). Those habitats are seldom processed by agricultural machinery, which may also increase the survival of leverets (*Weber, Roth & Kohli, 2019*). Furthermore, to mitigate predation risk, especially from mammalian predators, increased areas of shelter habitats with high structural richness should be preferred over homogeneous and smaller habitats (*Hummel et al., 2017*; *Laux, Waltert & Gottschalk, 2022*), due to lower predation risk in larger habitat patches within smaller proportion of habitat edges (*Laux, Waltert & Gottschalk, 2022*) and lower penetration of mammalian predators into habitats with higher and diverse vegetation cover (*Hummel et al., 2017*). Finally, despite some previous studies indicating that predator control can locally decrease predator density and thus predation risk for European hares and other prey species (*Reynolds & Tapper, 1995*; *Panek, Kamieniarz & Bresinski, 2006*), other studies demonstrated that such effects are likely temporary, ineffective in reducing predation risk or both, and, therefore, controversial (*Newsome, Crowther & Dickman, 2014*; *Baker & Harris, 2006*; *Kämmerle, Niekrenz & Storch, 2019*). Thus, we believe that improvements in habitat quality of arable landscapes by increasing the proportion and quality of extensively used non-farmed arable land may be more effective for reducing predation risk on leverets

and increasing European hare population growth than predator control programs (see also *Weber, Roth & Kohli, 2019*; *Schai-Braun et al., 2020*).

## ACKNOWLEDGEMENTS

We would like to thank Stanislav Grill for the map preparation and three reviewers who substantially improved the quality of the manuscript.

### Funding

This work was supported by the Czech Academy of Sciences in frame of the program Strategy AV 21, the research aim of the Czech Academy of Sciences (RVO 68081766), by the Technological Agency of the Czech Republic (SS05010238) and by the Internal Grant Agency of the Czech University of Life Sciences Prague (A_24_22). All external funding sources are declared here. There was no additional external funding received for this study. The funders had no role in study design, data collection and analysis, decision to publish, or preparation of the manuscript.

### Grant Disclosures

The following grant information was disclosed by the authors:
Czech Academy of Sciences in frame of the program Strategy AV 21, the research aim of the Czech Academy of Sciences: RVO 68081766.
Technological Agency of the Czech Republic: SS05010238.
Internal Grant Agency of the Czech University of Life Sciences Prague: A_24_22.

### Competing Interests

The authors declare there are no competing interests.

### Author Contributions

- Jan Cukor conceived and designed the experiments, performed the experiments, authored or reviewed drafts of the article, and approved the final draft.
- Jan Riegert analyzed the data, prepared figures and/or tables, authored or reviewed drafts of the article, and approved the final draft.
- Aleksandra Krivopalova analyzed the data, authored or reviewed drafts of the article, and approved the final draft.
- Zdeněk Vacek performed the experiments, authored or reviewed drafts of the article, and approved the final draft.
- Martin Šálek conceived and designed the experiments, performed the experiments, prepared figures and/or tables, authored or reviewed drafts of the article, and approved the final draft.

### Animal Ethics

The following information was supplied relating to ethical approvals (*i.e.*, approving body and any reference numbers):
The Czech Ministry of Environment approved the study (63479/2016-MZE-17214).

## Data Availability

The raw database of predation attempts are available in the Supplementary File.

## Supplemental Information

Supplemental information for this article can be found online at http://dx.doi.org/10.7717/peerj.17235#supplemental-information.

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
