# Peer review of "The low survival rate of European hare leverets in arable farmland: evidence from the predation experiment"

_PeerJ, doi:10.7717/peerj.17235_

## Round 0.1 · original submission · Major Revisions

Reviewer 3 likes the manuscript as is, and I agree that it is interesting and has value. Both Reviewers 1 and 2 have some doubts about whether you could justify your interpretations more explicitly. This is important to attend to, as a better explanation for your interpretation, or more caveats where necessary, will improve any applications of the results. Reviewer 1 also has very extensive and detailed comments. Please answer these to your best ability.

Reviewer 1 ·

Basic reporting

Please see attached pdf.

Experimental design

Please see attached pdf.

Validity of the findings

Please see attached pdf.

Annotated reviews are not available for download in order to protect the identity of reviewers who chose to remain anonymous.

·

Basic reporting

This study investigates the survival rate of leverets in Central European arable farmland, using hare dummies. The authors found that over 14 days, 22.2% of leverets survived, with predation events occurring mainly in the first six days. Mammalian predators and avian predators were responsible for most predation, with red foxes being the dominant predator.
I appreciate the study for its concise description, clear language, and adherence to the standards of PeerJ. My primary concerns are regarding the authors' interpretation of the results and whether the management actions they propose can indeed be derived from the presented results. Below, I will try to explain my concerns in more detail. Nonetheless, I believe that the results presented in this study are interesting and suitable for publication if the authors decide to be more cautious in interpreting the results.

Experimental design

The main aim of this study was to estimate European hare leveret survival (l. 81-90). However, the estimated survival rates were based on simulated leveret dummies, which may not reflect the actual rate of natural predation. Although this limitation was acknowledged by the authors (l. 87-88, 253-256), they nevertheless conclude based on the predation rate on the simulated leveret dummies that the true predation rate must be very low (title, lines 28, 167, 212). Unfortunately, however, the authors do not provide arguments how this conclusion can be justified from the predation rate on hare dummies. Furthermore, I am also not sure why a predation rate of 22% is considered as very low. It would be nice if the authors could argue at what point a predation rate becomes unsustainable.

Validity of the findings

On lines 270-285 the authors present conservation implications that in my opinion can not be derived from the results of the study. For example, the authors suggest that management actions should be implemented that promote habitat heterogeneity in extensively used arable land. This would imply a study that compares predation rate between heterogenous and homogenous habitats, which was not the case in the presented study.
Presentation of results from GLM (l. 193-196): Please provide the effect sizes of how vegetation cover and height affect predation rate. Currently, it is not possible to know whether predation rate is increasing or decreasing with vegetation cover and height, and it is also not possible to judge whether the effects are biologically relevant.

Additional comments

l. 109-110: You mean that each dummy remained 14 days in the field?
l. 216-221: All predation attempts were recorded during the first six days of the experiment. Could it not also be because the scent of the domestic rabbit urine faded over time?

Reviewer 3 ·

Basic reporting

This study uses dummy hare leverets to collect information about predation. I have found this manuscript well written, interesting and widening our knowledge. Studies with such management implications are really valuable, because they contribute in practice to conservation.
One improvement will be a better clarification of “use intensity” and “visit frequency” in Methods and a short explanation again in Table1.
Moreover, I think that some information about predation control efforts in country should be provided in Introduction and if are applied in Study Area. Also, a comment about the importance of these efforts in Conservation Implications will be useful.

Experimental design

Sufficient

Validity of the findings

high validity

---

## Round 0.2 · Minor Revisions

You have adequately addressed the main comments by the reviewers and the manuscript is much easier to understand now. The manuscript is now almost suitable for publication, but first can you please clarify the question about the sample sizes reported, and the other small comments by the reviewer?

·

Basic reporting

In this study, the authors investigated leveret survival in Central European arable farmland. They found that survival rate of the leveret dummies was only 22.2%, with predation events occurring only during the first six of fourteen days. Mammalian and avian predators accounted for most predation, with the red fox being the dominant predator.
I like to thank the authors for the revised version of the manuscript. They have put significant effort to incorporate the feedback from the reviewers. In my opinion, the manuscript has significantly improved.
This is a well conducted study, with a concise description of methods and results, clear language, and adherence to the standards of PeerJ.

Experimental design

Based on the description of the methods, I could not understand the provided sample sizes. On l. 132 you write that 48 independent plots were monitored, on l. 169 you state n = 46 for the analysis of survival rates, and on l. 222 the sample size for the test of the effect of the environmental variables was n = 127.

Validity of the findings

No comment.

Additional comments

- l. 26: the main evidence COMES mainly
- l. 56-60: Long sentence. I do not understand the last part of the sentence starting from "into remaining..."
- l. 95: remove "within the studied plots"
- l. 199-200: Sentence not clear. Please rewrite.
- l. 201-202: Please rewrite. I think the predation rate was highest during the first two days ("predation rate in the first two days increased" seems not accurate to me). It is also not clear to me what you mean with "moderate".
- l. 206: What is "mean use-intensity"?
- l. 332: add space to nighttimein
- Figure 1: Restrict y-axes at 1 as values >1 are not possible.

---

## Round 0.3 · Minor Revisions

I think two points require further clarification. Your explanation to the Reviewer of why the Ns differ is clear, but I think it is still possible to be confused when reading. At the beginning of the Results section, can you state something like "In the 46 plots, we observed 48 first-time predation events and 127 total predation events." Secondly, unfortunately your revision of the first lines of the Results section has added some confusion. You state that the survival rate was low, just 22% of dummies survived, implying that 78% of dummies were predated, then you say "All dummies were depredated within the first six days... Surprisingly, the rest of the dummies... " If 100% of dummies were predated, then "the rest" is 0%. This also does not match up with 78% and 22%. My impression is that you want to say "Survival rate of dummies was low with 78% of dummies being predated within the first 6 days. Surprising, the remaining 22% of dummies..." Is that correct? Please adjust and change as necessary.

---

## Round 0.4 · accepted · Accept

Thank you for making the last small changes. The paper is now ready for publication.